# Diabetes as a risk factor for the onset of frozen shoulder: a systematic review and meta-analysis

Brett Paul Dyer ⬤ , Trishna Rathod-Mistry, Claire Burton ⬤ , Danielle van der Windt ⬤ , Milica Bucknall

Primary Care Centre Versus Arthritis, School of Medicine, Keele University, Newcastle-under-Lyme, UK

**Correspondence to**
Dr Brett Paul Dyer;
b.p.dyer@keele.ac.uk

## ABSTRACT

**Objective** Summarise longitudinal observational studies to determine whether diabetes (types 1 and 2) is a risk factor for frozen shoulder.

**Design** Systematic review and meta-analysis.

**Data sources** MEDLINE, Embase, AMED, PsycINFO, Web of Science Core Collection, CINAHL, Epistemonikos, Trip, PEDro, OpenGrey and The Grey Literature Report were searched on January 2019 and updated in June 2021. Reference screening and emailing professional contacts were also used.

**Eligibility criteria** Longitudinal observational studies that estimated the association between diabetes and developing frozen shoulder.

**Data extraction and synthesis** Data extraction was completed by one reviewer and independently checked by another using a predefined extraction sheet. Risk of bias was judged using the Quality In Prognosis Studies tool. For studies providing sufficient data, random-effects meta-analysis was used to derive summary estimates of the association between diabetes and the onset of frozen shoulder.

**Results** A meta-analysis of six case–control studies including 5388 people estimated the odds of developing frozen shoulder for people with diabetes to be 3.69 (95% CI 2.99 to 4.56) times the odds for people without diabetes. Two cohort studies were identified, both suggesting diabetes was associated with frozen shoulder, with HRs of 1.32 (95% CI 1.22 to 1.42) and 1.67 (95% CI 1.46 to 1.91). Risk of bias was judged as high in seven studies and moderate in one study.

**Conclusion** People with diabetes are more likely to develop frozen shoulder. Risk of unmeasured confounding was the main limitation of this systematic review. High-quality studies are needed to confirm the strength of, and understand reasons for, the association.

**PROSPERO registration number** CRD42019122963.

## STRENGTHS AND LIMITATIONS OF THIS STUDY

⇒ This systematic review is the first to summarise the results of studies estimating the longitudinal association between diabetes and the onset of frozen shoulder.

⇒ Robust meta-analytical methods were used to synthesise and analyse data.

⇒ Sensitivity to influential estimates and sensitivity to small study bias were assessed.

⇒ Risk of bias was judged to be high in seven studies and moderate in one study; this limits the certainty in evidence.

⇒ Only two cohort studies were identified, which meant that pooling of association estimates was not suitable.

## INTRODUCTION

Frozen shoulder, also known as adhesive capsulitis, is a painful and severely debilitating condition. The inflammatory contracture of the glenohumeral joint capsule in frozen shoulder restricts both active and passive range of motion, with loss of external rotation being especially characteristic of this condition.[1]

Frozen shoulder generally presents between the ages of 50 years and 60 years and rarely presents before 40 years.[2] Women (58%) are more likely to develop frozen shoulder than men (42%).[3] The contralateral shoulder is also affected in 6%–17% of patients.[4] Although the exact aetiology remains unclear, several factors have been found to be associated with frozen shoulder, including trauma,[3] thyroid dysfunction,[5–7] cardiovascular disease,[2 8] metabolic factors[7 9–11] and other musculoskeletal conditions such as Dupuytren's contracture.[12 13] The most common comorbidity in people with frozen shoulder is diabetes,[2] both type 1 and type 2.[6] The prevalence of frozen shoulder in the general population is around 0.75%,[1] but the prevalence of frozen shoulder in people with diabetes is much higher. A meta-analysis of 13 cross-sectional studies estimated the prevalence of frozen shoulder in populations with diabetes to be 13.4% (95% CI 10.2% to 17.2%).[14]

Diabetes is a term used to describe a group of chronic diseases characterised by hyperglycaemia. The two most prevalent types of diabetes are type 1 and type 2, making up 8% and 90% of cases, respectively.[15] It is well known that people with diabetes are at risk of

complications such as cardiovascular disease, retinopathy, neuropathy and nephropathy,[16] although the musculo-skeletal complications of diabetes are not as well known.[17] Musculoskeletal conditions, such as frozen shoulder, can significantly affect the quality of a patient's life and should not be overlooked. Our previous systematic review and narrative synthesis of 28 studies has shown that patients with diabetes may experience worse outcomes from frozen shoulder than people without frozen shoulder.[18]

It has been suggested that diabetes may be a cause of frozen shoulder through glycation processes and/or inflammatory processes leading to capsular fibrosis and subsequent contracture.[7 19 20] To understand whether diabetes could potentially be a cause of frozen shoulder, it is necessary (although not sufficient) to have evidence of the temporal relationship between diabetes and frozen shoulder.[21] This systematic review aims to summarise evidence from longitudinal observational studies to understand the temporal relationship between diabetes and frozen shoulder.

## METHODS
### Search strategy
The protocol for this systematic review was registered on PROSPERO (CRD42019122963), and the review was conducted and reported using Preferred Reporting Items for Systematic Reviews and Meta-Analyses guidelines.[22] A systematic literature search of MEDLINE, Embase, AMED, PsycINFO, Web of Science Core Collection, CINAHL, Epistemonikos, Trip, PEDro, OpenGrey and The Grey Literature Report was carried out in January 2019 and updated in June 2021. Reference lists of eligible studies were screened. Additionally, a professional contact of one author (DvdW) was contacted to identify further studies. We retrieved all epidemiological studies containing index terms (eg, Medical Subject Headings) and free-text words related to diabetes and shoulder pain more generally (not limited to frozen shoulder) to reduce the risk of missing potentially relevant publications. The search strategy for MEDLINE, which was constructed with the support of a health information specialist, can be found in online supplemental appendix A.

### Study selection
Reviewer BPD screened all titles and abstracts to check eligibility using the predefined inclusion and exclusion criteria, and reviewers MB and CB independently checked a 20% random sample. Reviewer BPD checked all full-texts for eligibility using the inclusion and exclusion criteria, and reviewers MB, CB and TR also independently checked eligibility. Disagreements regarding the inclusion of studies were resolved through discussion with DvdW.

### Inclusion criteria
To be eligible for inclusion, studies were required to have a longitudinal, prospective or retrospective, observational study design. Cohort studies were required to have a study population consisting of people without frozen shoulder at inclusion and must have established whether diabetes was present at baseline (all types of diabetes were considered). Case–control studies were required to have a study population consisting of people with frozen shoulder and a control group without frozen shoulder, with diabetes defined as the exposure of interest. The paper must have presented an OR, risk ratio or HR, or they must have presented sufficient data to allow the associations to be estimated. There were no restrictions to setting; population-based as well as clinical cohorts were eligible. All non-English language papers were assessed by reviewers with appropriate language skills. Cross-sectional studies and case series were excluded. Studies were also excluded if a full text could not be obtained.

### Data extraction and risk of bias
Data extraction was completed by reviewer BPD and was independently checked by reviewers MB and TR. Types of data extracted included details of study design, setting, sample characteristics, exposure/outcome/covariate measurement, inclusion and exclusion criteria, sample size, attrition, covariate conditioning, follow-up time, statistical analysis, association estimates (OR, risk ratio or HR) or raw data to estimate association sizes if they were not already presented. Risk of bias was independently assessed by pairs of reviewers (BPD, MB and TR). Risk of bias was judged using the Quality In Prognosis Studies (QUIPS) tool.[23] The QUIPS tool covers six domains: (1) study participation, (2) study attrition, (3) prognostic/risk factor measurement, (4) outcome measurement, (5) study confounding and (6) statistical analysis and reporting. Each of the six domains is scored as being at a low, medium or high risk of bias.[23] Domain scores were used to guide judgement of the overall risk of bias (scored as low, medium or high) for the study. Overall risk of bias was based on author judgement, and the use of a tallied or summated score was avoided. All disagreements regarding data extraction and assessment of risk of bias were resolved by discussion.

### Data analysis
Case–control studies and cohort studies were analysed separately. Narrative synthesis was used where less than five studies were present and a random-effects meta-analysis model was used to calculate a summary estimate when five or more studies were present. Cohort study associations were measured using hazard ratios and case–control study associations were estimated using ORs. Where adjusted and crude estimates were both presented, the adjusted estimate was used. Where a zero-cell count was present within the results of a study, a continuity correction of 0.5 was added to all cells for that study. Restricted maximum likelihood estimation[24] was used to estimate the between-study variance, $\tau^2$, and the Hartung-Knapp-Sidik-Jonkman variance correction method[25] was used in the estimation of the pooled effect CI. Heterogeneity

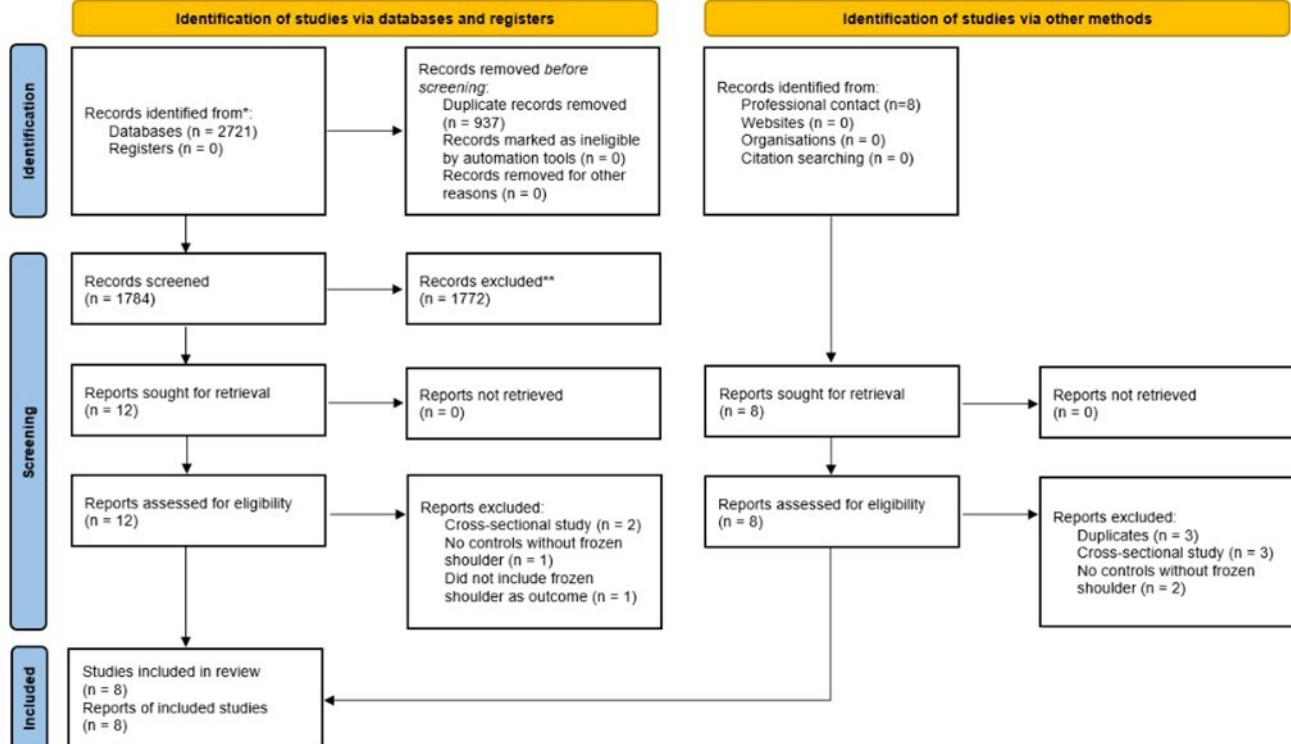

**Figure 1** PRISMA flow diagram summarising record identification and study selection. PRISMA, Preferred Reporting Items for Systematic Reviews and Meta-Analyses.

was assessed using Cochran's Q statistic, complemented by the $I^2$ index.[26] Prediction intervals were not estimated since they are inaccurate when there is little heterogeneity ($I^2 < 0.3$), or an imbalance in study sizes exists, both of which were found in the meta-analysis in this review (see Section 3).[27] A forest plot was used to visualise results of individual results and of the pooled estimate. Evidence of small-study bias was assessed with a funnel plot of log ORs against their standard errors.[28] A test for funnel plot asymmetry was not used since the meta-analysis included less than ten studies.[29] The influence of each study on the overall pooled estimate was assessed by repeating the meta-analysis, each time leaving out a single study.[30] Statistical analysis was carried out using Stata V.16.1.[31]

**Patient and public involvement**
No patient involved.

**RESULTS**
The searches identified 1784 unique citations, 12 of which were selected for full-text screening, and 8 studies consisting of a total of 346 278 people fulfilled the inclusion criteria (figure 1). Table 1 summarises information on risk of bias, study design, setting, participants, sample size and methods used for diagnosing diabetes and frozen shoulder. Of the eight studies that met the criteria for inclusion, six[32–37] had case–control designs and two[38 39] had cohort designs. Three studies[37–39] (including the two cohort studies) collected information from electronic

health records (EHRs); four studies[33–36] were hospital-based, and one study[32] was based in a physical therapy clinic. Among the case–control studies, the percentage of female cases ranged from 52% to 75%, and the mean age for cases ranged from 52.8 years to 57.2 years.

Presence of diabetes was identified using ICD-9 codes (codes to classify diseases, symptoms, clinical findings and causes of disease and injury) from electronic health records in three studies,[37–39] self-reported in three studies,[32 33 36] identified with a glucose test or if the patient was receiving drug treatment for diabetes in one study[35] and was unclear in one study.[34] Frozen shoulder was identified using[37–39] ICD-9 codes in three studies and was diagnosed clinically in five studies.[32–36] Only one study[39] reported the types of diabetes that the participants had. Lo *et al*[39] stated that 296 (5.8%) of the 5109 people with diabetes in their study had type 1 diabetes. Two studies were conducted in Taiwan[38 39]; two were conducted in the USA[32 37]; and the remaining four were conducted in China,[33] South Korea,[34] Israel[35] and Australia.[36]

Overall QUIPS risk of bias scores for each study can be found in table 1, and full QUIPS assessments can be found in table 2. Overall, there was a 75% agreement between reviewers across the individual bias domains, and reviewers agreed on four of the eight overall risk of bias scores. One of the cohort studies[39] was scored as being at a moderate risk of bias for their overall study rating, and the other seven studies were rated as being at a high risk of bias overall. A bar graph of the scores for

**Table 1** Characteristics of studies on diabetes as a risk factor for frozen shoulder

| Source | Risk of bias (QUIPS, overall assessment) | Design and setting | % Female | Mean age (years) | Sample size | Method to diagnose diabetes and frozen shoulder | Variables conditioned on |
|---|---|---|---|---|---|---|---|
| Case–control studies | | | | | | | |
| Boyle-Walker et al[32] | High | Sex-matched case–control at physical therapy clinic in the USA | Case group: 75%, control group: 68% | Not reported | Cases: 32, controls: 31 | Diabetes: self-reported questionnaire Frozen shoulder: clinically diagnosed | Sex-matched |
| Li et al[33] | High | Hospital-based case–control matched on time of hospitalisation in China | Case group: 63%, control group: 55% | Cases: 57.2, controls: 45.9 | Cases: 182, controls: 196 | Diabetes: face-to-face interview Frozen shoulder: clinically diagnosed | Matched on time of hospitalisation, adjusted for history of minor shoulder trauma |
| Lee et al[34] | High | Hospital based age-matched and sex-matched case–control in South Korea | Case group: 55%, control group: not reported | Cases: 52.8, controls: not reported | Cases: 40, controls: 40 | Diabetes: unclear Frozen shoulder: clinically diagnosed | Age-matched and sex-matched |
| Milgrom et al[35] | High | Hospital based age-matched case–control in Israel | Case group: 60%, control group: 65% | Cases: 54.9, controls: 55.4 | Cases: 126, controls: 98 | Diabetes: If patient was receiving drug treatment for diabetes or whose serum glucose was higher than 200 mg/dL Frozen shoulder: clinically diagnosed | Age-matched |
| Wang et al[36] | High | Hospital based age-matched and sex-matched case–control in Australia | Case group: 64%, control group: 58% | Cases: 56, controls: 55.3 | Cases: 87, controls: 176 | Diabetes: self-reported Frozen shoulder: clinically diagnosed | Age-matched and sex-matched |
| Kingston et al[37] | High | Sex-matched case–control using EHRs in the USA | Case group: 58%, control group: 58% | Cases: 56.4, controls: not reported | Cases: 2190, controls: 2190 | Diabetes: ICD-9 code Frozen shoulder: ICD-9 code | Sex-matched |
| Cohort studies | | | | | | | |
| Huang et al[38] | High | Age-matched and sex-matched cohort with 3-year follow-up using electronic health records in Taiwan | Exposed group: 47%, non-exposed group: 47% | Exposed group: 55.7, non-exposed group: 55.5 | Exposed group: 78 827, non-exposed group: 236 481 | Diabetes: ICD-9 code Frozen shoulder: ICD-9 code | Age-matched and sex-matched, multivariable analysis adjusted for age, sex and dyslipidaemia |

Continued

**Table 1** Continued

| Source | Risk of bias (QUIPS, overall assessment) | Design and setting | % Female | Mean age (years) | Sample size | Method to diagnose diabetes and frozen shoulder | Variables conditioned on |
|---|---|---|---|---|---|---|---|
| Lo et al[39] | Moderate | Cohort with 8-year follow-up using EHRs in Taiwan | Exposed group: 52%, non-exposed group: 51% | Not reported | Exposed group: 5109, non-exposed group: 20 473 | Diabetes: ICD-9 code Frozen shoulder: ICD-9 code | Multivariable analysis adjusted for age, income, stroke, hypertension, hyperlipidaemia, obesity and chronic obstructive pulmonary disease |

EHR, electronic health record; ICD-9, International Classification of Diseases, Ninth Revision; QUIPS, Quality In Prognosis Studies.

individual risk of bias domains can be found in online supplemental appendix figure B1. Risk of bias was generally high across most domains, but especially so for the risk of unaccounted confounding, which was scored as being at a high risk of bias in all eight studies. Five of the case–control studies[32 34–37] only accounted for age, gender or a combination of the two. One study[33] matched on the time of hospitalisation and adjusted for history of minor shoulder trauma. One cohort study[38] adjusted for age, sex and dyslipidaemia; the other cohort study[39] adjusted for age, income, stroke, hypertension, hyperlipidaemia, obesity and chronic obstructive pulmonary disease.

Six case–control studies including a total of 5388 people were pooled in a random-effects meta-analysis, with a pooled OR of 3.69 (95% CI 2.99 to 4.56) (figure 2). The raw data extracted from each study that was used to calculate ORs can be found in online supplemental

appendix table C1. The estimated between-study variance was small ($\tau^2$<0.01, 95% CI <0.01 to 0.23), and little heterogeneity was detected (Q=2.07, df=5, p=0.84; $I^2$<0.01%, 95% CI <0.1% to 67.6%), but the estimate for $I^2$ was imprecise as indicated by the wide 95% CI. The influence analysis showed that excluding the largest study,[37] which contained 4380 of the 5388 participants, greatly reduced the precision of the pooled estimate but did not substantially affect the value of the pooled estimate (figure 3). Further, excluding any other single study did not substantially affect the value of the pooled estimate (figure 3). The two studies with the smallest SEs for their effect estimates had the largest ORs, making the funnel plot appear unsymmetrical. However, due to the small number of studies contributing to the funnel plot, the asymmetrical appearance could be due to chance (figure 4).

**Table 2** QUIPS domain scores for each primary study

| Source | Participation | Study attrition | Risk factor measurement | Outcome measurement | Confounding | Statistical analysis and presentation | Overall risk of bias |
|---|---|---|---|---|---|---|---|
| Case–control studies | | | | | | | |
| Boyle-Walker et al[32] | High | Moderate | High | Moderate | High | Moderate | High |
| Li et al[33] | Moderate | Low | Moderate | High | High | High | High |
| Lee et al[34] | Moderate | Low | Moderate | Moderate | High | Moderate | High |
| Milgrom et al[35] | Moderate | Low | Low | Low | High | Low | High |
| Wang et al[36] | Low | Low | Low | Low | High | Low | High |
| Kingston et al[37] | Low | Moderate | Moderate | Low | High | Moderate | High |
| Cohort studies | | | | | | | |
| Huang et al[38] | Low | Moderate | Low | High | High | High | High |
| Lo et al[39] | Low | Low | Low | Moderate | High | Low | Moderate |

QUIPS, Quality In Prognosis Studies.

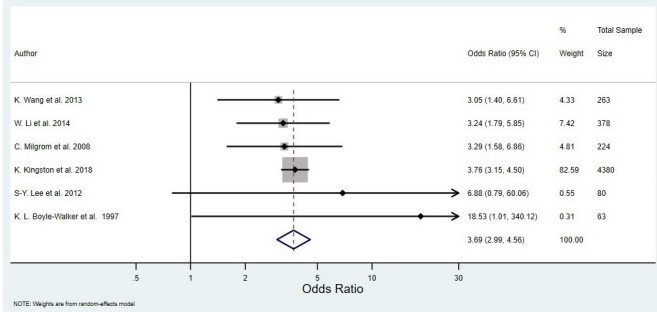

**Figure 2** Random effects meta-analysis of the association between diabetes and the odds of developing frozen shoulder.

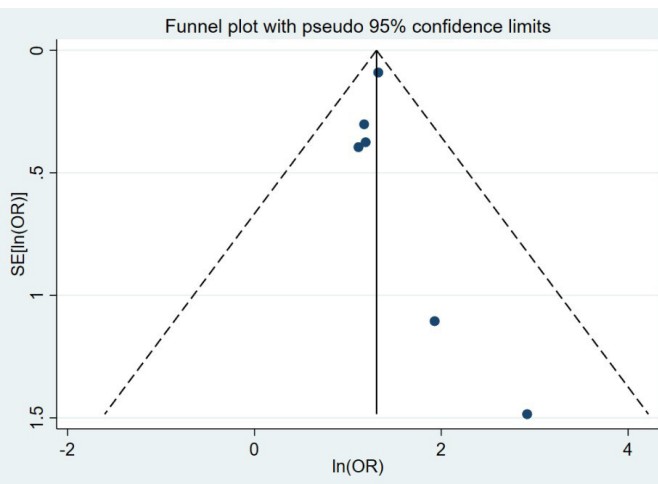

**Figure 4** Funnel plot of log ORs for developing frozen shoulder in people with diabetes versus those without diabetes.

The two cohort studies that were identified used Cox proportional hazards models and obtained results suggesting that people with diabetes were more at risk of developing frozen shoulder. One cohort study[38] using electronic health records from Taiwan, with a 3-year follow-up and consisting of 315 308 people reported an age-adjusted, sex-adjusted and dyslipidaemia-adjusted HR of 1.32 (95% CI 1.22 to 1.42). Another cohort study,[39] with an 8-year follow-up, consisting of 25 582 people, also using electronic health records from Taiwan, estimated an age-adjusted, income-adjusted, stroke-adjusted, hypertension-adjusted, hyperlipidaemia-adjusted, obesity-adjusted and chronic obstructive pulmonary disease-adjusted HR of 1.67 (95% CI 1.46 to 1.91).

## DISCUSSION

This systematic review aimed to summarise evidence from longitudinal observational studies to determine whether diabetes (types 1 and 2) is a risk factor for frozen shoulder.

Eight studies met the eligibility criteria for the review; each individual study demonstrated evidence to suggest that diabetes is associated with the onset of frozen shoulder. Our meta-analysis of six case–control studies yielded a pooled OR of 3.69 (95% CI 2.99 to 4.56), and the value of the pooled estimate was robust to the omission of any individual study. The OR estimates of all but one study[37] were imprecise with large CIs; this meant that the CIs overlapped well, resulting in a small $I^2$ value. It is also important to note that Cochran's Q statistic should

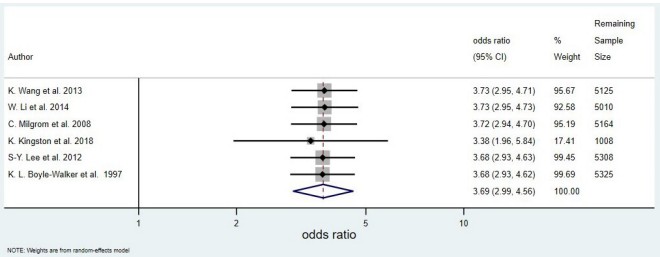

**Figure 3** Influence plot showing the result of repeating the original meta-analysis (figure 2), each time with a different primary study removed.

be interpreted with caution since the number of studies included in the analysis was small.[40]

The funnel plot was unsymmetrical. However, given that a small number of studies were available, it was difficult to assess accurately whether any small-study bias was present or if the appearance was due to chance. Since our influence analysis has shown that the inclusion/exclusion of any individual study had very little impact on the pooled effect estimate, any potential small-study bias would be unlikely to substantially affect the results.

Two cohort studies were identified, both of which corroborate the evidence from the six case–control studies reported previously, that people with diabetes are more likely to develop frozen shoulder than those without diabetes. Of the two cohort studies, one was deemed to be at a high risk of bias and the other at a moderate risk of bias. The HRs in the two studies did differ, which could have partly been due to the differences in the covariates that were adjusted for and/or the differences in the duration of follow-up. Both studies were rated as being at a high of bias for the outcome-measurement domain as the length of follow-up (3[38] and 8 years[39]) was deemed too short to establish whether a patient would develop frozen shoulder in the future. Previous studies have suggested that the duration of diabetes may be associated with the risk of developing frozen shoulder,[41 42] with one of the cohort studies in this review also stating that their study suggested that 'the development of (frozen shoulder) is associated with the duration of diabetes'.[38] Therefore, future studies should ensure that the follow-up period is long enough to observe participants from diabetes diagnosis through to the ages for which frozen shoulder is common. A cross-sectional study of 1373 patients presenting with frozen shoulder estimated that the mean age of onset for frozen shoulder was 55.4 years with an SD of 9.9 years.[3]

The following three paragraphs describe some limitations that may complicate the understanding of the

association between diabetes and the onset of frozen shoulder.

The two cohort studies in the review were both conducted using EHRs. EHR datasets can provide large sample sizes with long follow-up periods and detailed patient medical record history.[43] Misdiagnosis and miscoding in EHRs are common limitations and could potentially result in a risk of bias for frozen shoulder measurement.[44] Research in the UK[45] and in the Netherlands[46] has shown that general practitioners often use non-specific shoulder pain codes instead of codes for specific shoulder conditions, for example, frozen shoulder. This would lead to an underdiagnosis of frozen shoulder. Further, this misclassification may be differential since clinicians may feel more confident in providing a specific frozen shoulder diagnosis in patients with diabetes due to the pre-existing knowledge of the association between the two conditions. Conversely, it has also been noted that frozen shoulder is sometimes used as a 'waste-bin diagnosis' for patients presenting with any stiff and painful shoulder.[47] Thus, EHR data may include other shoulder conditions with similar clinical presentations being coded as frozen shoulder.

Another important limitation was the overall poor adjustment for confounding variables. All eight studies were rated as being at a high risk of unaccounted confounding. In each study, confounders were either ignored[32 34–38] or inappropriate statistical methods, such as univariable prefiltering and stepwise selection, were used.[33 38 39] These methods are especially poorly suited for aetiological models.[48] Thus, these studies may have missed potentially important confounders[33 38 39] or erroneously adjusted for mediators, such as stroke.[39]

The systematic review is also limited by there being only two cohort studies, meaning that pooling association estimates was not possible. Cohort studies are particularly useful for gaining a better understanding of temporal associations, as this review aimed to do. Further, both cohort studies were conducted in Taiwan using existing data from EHRs. Future studies with prospective designs will help to gauge whether the findings of these two cohort studies are reproducible and whether the results are consistent across different populations.

Previously, a meta-analysis of cross-sectional studies established that frozen shoulder was more prevalent in people with diabetes than among people without diabetes. This systematic review provides evidence of a temporal relationship between diabetes and frozen shoulder. Understanding the temporal relationship is key to explaining why diabetes and frozen shoulder are associated; however, further high-quality research with appropriate methods and study design is required to confirm the strength of the association and establish whether diabetes is indeed a cause of frozen shoulder.

While sound and reliable epidemiological evidence of a causal relationship between diabetes and frozen shoulder is currently unavailable, elsewhere in the literature, researchers have hypothesised about potential pathological mechanisms through which diabetes may lead to frozen shoulder. Current evidence, based on histological studies, suggests that a pathophysiological process consisting of chronic inflammation and capsular fibrosis leads to the contracture in frozen shoulder.[49 50] It has been hypothesised that the accumulation of advanced glycation end products (AGEs), which lead to the cross-linking of collagen,[51 52] may explain the fibrosis in the capsule of patients with frozen shoulder.[33] Glycation is a process by which simple sugars bond to proteins, which is enhanced by persistent hyperglycaemia. Thus, the role of glycation and AGEs in the fibrosis of the shoulder capsule could potentially be a reason why diabetes is associated with frozen shoulder. Another potential reason why diabetes may be associated with frozen shoulder is that hyperglycaemia may induce proinflammatory cytokines[53] which have been found to be elevated in the capsule and synovium of patients with frozen shoulder.[54]

The association between glycaemic control and the risk of developing frozen shoulder should also be a focus for future research. One study found evidence to suggest that poor long-term glycaemic control in people with diabetes is associated with an increased incidence of frozen shoulder,[55] while another study found no association between HbA1c level in people with diabetes and the prevalence of frozen shoulder.[56] Further research is required to investigate whether glycaemic control is associated with the development of frozen shoulder.

## CONCLUSION

In summary, people with diabetes are more at risk of developing frozen shoulder than people without diabetes. However, existing research is limited by the high risk of unmeasured confounding. To better understand the nature of the relationship between diabetes and the onset of frozen shoulder, it is necessary to have high-quality cohort studies that use causal inference methods that are appropriate for aetiological modelling. Given the existing evidence that has been summarised in this review, clinicians should consider checking whether patients with diabetes are experiencing shoulder pain at their routine follow-up appointments. An early diagnosis will help the clinician to provide treatment for the pain and lack of function that result from frozen shoulder.

**Contributors** Data extraction and risk of bias assessment were performed by BPD, MB and TR. BPD performed the meta-analysis and narrative synthesis of the results and drafted the initial manuscript. All authors contributed to the conception of the study and systematic review of the study selection, editing and approval of the final manuscript. BPD is the guarantor.

**Funding** This study was supported by the Versus Arthritis PhD scholarship scheme (grant number 21899). The authors also thank the information specialists of evidence synthesis team at the Primary Care Centre Versus Arthritis for their advice regarding the search strategy and Elaine Willmore for taking time to check the results of our searches for this review. Further thanks are given to Dr Linda Chesterton, who was involved in the initial conceptualisation of this study. CB is funded by an National Institute for Health Research Clinical Lectureship.

**Disclaimer** The views expressed are those of the authors and not necessarily those of the National Health Service, the NIHR, or the Department of Health and Social Care.

**Competing interests** None declared.

**Patient and public involvement** Patients and/or the public were not involved in the design, conduct, reporting or dissemination plans of this research.

**Patient consent for publication** Not applicable.

**Ethics approval** Not applicable.

**Provenance and peer review** Not commissioned; externally peer reviewed.

**Data availability statement** All data relevant to the study are included in the article or uploaded as supplementary information. Not applicable. (Data have been included in table 1 and online supplemental appendix table C1).

**ORCID iDs**
Brett Paul Dyer http://orcid.org/0000-0001-8039-7631
Claire Burton http://orcid.org/0000-0003-4688-3075
Danielle van der Windt http://orcid.org/0000-0002-7248-6703

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
