## [Reviewer comments · BMJ Open]

ARTICLE DETAILS

TITLE (PROVISIONAL)	Diabetes as a Risk Factor for the Onset of Frozen Shoulder: A Systematic Review and Meta-Analysis
AUTHORS	Dyer, Brett; Rathod, Trishna; Burton, Claire; van der Windt, Danielle; Bucknall, Milica

VERSION 1 – REVIEW

REVIEWER	Dueñas, Lirios University of Valencia, Department of physical therapy
REVIEW RETURNED	18-Mar-2022

GENERAL COMMENTS	The authors have submitted a manuscript that summarise longitudinal observational studies to determine whether diabetes is a risk factor for the onset of frozen shoulder. Although the study is interesting, there are some issues that need to be addressed. General and specific comments are listed below. GENERAL COMMENTS. The study design and data treatment are adequate, but the manuscript presents some weak points. Additional editing is necessary. 1) The proposed title and your hypothesis are not linked. A risk-factor cannot determine cause and effect. A temporal relationship is necessary to determine whether the association between diabetes and frozen shoulder is causal, but this is not sufficient. Please, explain better this aspect and formulate a more suitable hypothesis according to this. I suggest rephrasing the hypothesis, abstract and conclusions taking this into account. 2) The introduction is not smooth or balanced. I miss some paragraph or data about diabetes, to present a more equilibrated Introduction section, and to better introduce further the aim of your study. 3) Limitations are not included in the discussion, which makes this section incomplete. 4) Conclusion paragraph is not well linked, and too generic sentences are used (specially the last one). 5) Some citations are not in brackets but between parentheses (author, year). Please, revise and use the same way for citations throughout the manuscript. SPECIFIC COMMENTS 1. Key words
---

	*I miss some words as adhesive capsulitis, risk factor, frozen shoulder,... and some other used are too generic (rheumatology or epidemiology) 2. Abstract *'Onset' word following 'frozen shoulder' is sometimes redundant. Please, avoid using 'onset' so many times, as using frozen shoulder is enough some of the times. 3. Strengths *Lines 39-43. Awkward wording. This sentence needs re-writing. 4. Introduction *Lines 5-8. Better specify what this heterogeneity is about, in order to better explain the rationale of your work. *Lines 8-9. Wrong way of citing, as Zeik is between brackets. Please, revise. * The rationale and hypotheses needs to be changed. Or at least, better explained. 5. Methods *Subheadings would allow a more fluent reading. Some subheadings such as search strategy, inclusion criteria, study selection, bias assessment, data analysis, etc., could be included. * Lines 44-45. It seems as if ALL cross-sectioned studies and case series found were studies where the full text was not available. And that this was the reason why they were excluded. Is this right? * Was any assessment of heterogeneity done? Why not, as you mention that heterogeneity was a limitation of the study that preceded yours (i.e., Zeik's) * Was the same reviewer the one who reviewed all non-English language papers? It sound strange. 6. Results * Page 7, line 3. 'Doing so' is a too informal term. The same happens with 'but specially so for...' * Please reference the studies when you mention them. For instance: 'in one study (REF).... Frozen shoulder was identified using ICD-9 codes in three studies (REF)... This point applies to all Results section. * Page 7 lines 27-29. Past tense should be used instead of present tense. 7. Discussion * Please, check the verb tenses. * Page 8, line 13. Are the italics necessary? * Figures: sometimes the thousands are separated by a comma and sometimes not: please, check it throughout the manuscript, including the tables as well. * Page 9, lines 13-15. Musculoskeletal pain is too vague for the conclusion, and you have not investigated this but frozen shoulder. 7. Figures * PRISMA flow diagram. Different letter sizes? (not sure) 8. Appendices * I would include Risk of Bias table in the manuscript instead of in the Appendix.
--	---

REVIEWER	Finestone, Aharon Tel Aviv University
REVIEW RETURNED	18-Apr-2022

GENERAL COMMENTS	This meta-analysis assesses the association between diabetes and frozen shoulder. I think the study was performed well in line with current guidelines for meta-analyses and do not have any major concerns.
--

	I am not sure I agree with the statement that the funnel plot shows only slight asymmetry. The fact that the two studies on the bottom right are way over to the right is clear asymmetry, and in my opinion points to a significant publication risk even with the influence analysis stated. I would prefer to say there were too few studies to infer, than "slight asymmetry". Therefore the authors might want to rephrase P7L56. I'm not sure about the English but does "asymmetry" in P7L56 agree with "unsymmetrical" in P7L32? I think the main problem not addressed in the discussion is the diagnosis of "frozen shoulder". As a co-author of one of the studies included [21] where every subject was examined and fulfilled the accurate definition of frozen shoulder, and as a clinician who sees patients with "frozen" shoulders frequently, I can guarantee we understand very little about the pathology. Obviously, some cases include an inflammatory process that has been reported in the subjects that have been biopsied. And the extreme cases that are "frozen" even under general anesthesia are definitely very hard to manage. In these cases, the other proposed mechanisms in P8L34-46 may be relevant. But the vast majority of subjects with "frozen shoulder" have no histopathological testing and have much milder signs and symptoms, which may not infer any intra-articular pathology. Many cases referred to me and my colleagues as "frozen shoulder" with that diagnosis encoded can be relieved in the clinic (at least temporarily) by various maneuvers. But they would be captured in "big data" studies like [24] & [25]. In the end, many of the cases are simply a clinical presentation, where the main manifestation is pain. In many, limited motion is probably secondary to the pain. I do not think the authors necessarily want to delve into pain theory. But I think that the study is based on diagnoses of problematic reliability, made by a group of physicians whose training in MSK medicine may not be the best. This should be stated as a serious limitation.
--	---

REVIEWER	Black, Deborah The University of Sydney, Faculty of Medicine and Health
REVIEW RETURNED	17-Aug-2022

GENERAL COMMENTS	This manuscript is well-written and presents results clearly. The authors point out that there was a lack of adjustment for confounders. What were the confounders? Does Table 1 give all the details of the confounders in the section "Variables conditioned on". In some papers that seems very minimal for observational papers. Some discussion about the quality of the journals that accepted these very likely biased papers would enhance the manuscript. Should the PRISM flowchat be presented in the body of the manuscript rather than the appendix?
--

VERSION 1 – AUTHOR RESPONSE

GENERAL COMMENTS.

The study design and data treatment are adequate, but the manuscript presents some weak points. Additional editing is necessary.

- 1) The proposed title and your hypothesis are not linked. A risk-factor cannot determine cause and effect. A temporal relationship is necessary to determine whether the association between diabetes and frozen shoulder is causal, but this is not sufficient. Please, explain better this aspect and formulate a more suitable hypothesis according to this. I suggest rephrasing the hypothesis, abstract and conclusions taking this into account.

I have re-written large parts of the introduction to better explain the motivation for the research and to explain the aim of the study.

The conclusion in the abstract states that “People with diabetes are more likely to develop frozen shoulder.” This is not a causal claim.

The conclusion in the main text also does not make a causal claim and explains that there is a high risk of unmeasured confounding. “In summary, people with diabetes are more at risk of developing frozen shoulder than people without diabetes. However, existing research is limited by the high risk of unmeasured confounding. To better understand the nature of the relationship between diabetes and the onset of frozen shoulder, it is necessary to have high-quality cohort studies that use covariate selection methods (causal inference methods) that are appropriate for aetiologic modelling.”

- 2) The introduction is not smooth or balanced. I miss some paragraph or data about diabetes, to present a more equilibrated Introduction section, and to better introduce further the aim of your study.

I have added the following paragraph.

“Diabetes is a term used to describe a group of chronic diseases characterised by hyperglycaemia. The two most prevalent types of diabetes are type 1 and type 2 diabetes, making up 8% and 90% of cases, respectively [15]. It is well-known that people with diabetes are at risk of complications such as cardiovascular disease, retinopathy, neuropathy, and nephropathy [16], although the musculoskeletal complications of diabetes are not as well-known [17]. Musculoskeletal conditions, such as frozen shoulder, can significantly affect the quality of a patient’s life and should not be overlooked. Our previous systematic review and narrative synthesis of 28 studies has shown that patients with diabetes may also experience worse outcomes from frozen shoulder than people without frozen shoulder [18].”

- 3) Limitations are not included in the discussion, which makes this section incomplete.

The limitation section in the discussion has been expanded upon.

“The two cohort studies in the review were both conducted using Electronic Health Records (EHRs). EHR datasets can provide large sample sizes with long follow-up periods and detailed patient medical record history [43]. Misdiagnosis and miscoding in EHRs are common limitations and could potentially result in a risk of bias for frozen shoulder measurement [44]. Research in the UK [45] and in the Netherlands [46] has shown that general practitioners often use non-specific shoulder pain codes instead of codes for specific shoulder conditions, e.g., frozen shoulder. This would lead to an underdiagnosis of frozen shoulder. Further, this misclassification may be differential since clinicians may feel more confident in providing a specific frozen shoulder diagnosis in patients with diabetes due to the pre-existing knowledge of the association between the two conditions. Conversely, it has also been noted that frozen shoulder is sometimes used as a “waste-bin diagnosis” for patients presenting with any stiff and painful shoulder [47]. Thus, EHR data may include other shoulder conditions with similar clinical presentations being coded as frozen shoulder.

Another important limitation was the overall poor adjustment for confounding variables. All eight studies were rated as being at a high risk of unaccounted confounding. In each study, confounders were either ignored [32,34-38] or inappropriate statistical methods, such as univariable prefiltering and

stepwise selection, were used [33,38,39]. These methods are especially poorly suited for aetiologic models [48]. Thus, these studies may have missed potentially important confounders [33,38,39] or erroneously adjusted for mediators, such as stroke [39].

The systematic review is also limited by there being only two cohort studies, meaning that pooling association estimates was not possible. Cohort studies are particularly useful for gaining a better understanding of temporal associations, as this review aimed to do. Further, both cohort studies were conducted in Taiwan using existing data from EHRs. Future studies with prospective designs will help to gauge whether the findings of these two cohort studies are reproducible, and whether the results are consistent across different populations.”

- 4) Conclusion paragraph is not well linked, and too generic sentences are used (specially the last one).

The conclusion now reads “In summary, people with diabetes are more at risk of developing frozen shoulder than people without diabetes. However, existing research is limited by the high risk of unmeasured confounding. To better understand the nature of the relationship between diabetes and the onset of frozen shoulder, it is necessary to have high-quality cohort studies that use causal inference methods that are appropriate for aetiologic modelling. Given the existing evidence that has been summarised in this review, clinicians should consider checking whether patients with diabetes are experiencing shoulder pain at their routine follow-up appointments. An early diagnosis will help the clinician to provide treatment for the pain and lack of function that result from frozen shoulder.”

- 5) Some citations are not in brackets but between parentheses (author, year). Please, revise and use the same way for citations throughout the manuscript.

I have now changed all citations to be in the same style.

SPECIFIC COMMENTS

1. Key words

*I miss some words as adhesive capsulitis, risk factor, frozen shoulder,... and some other used are too generic (rheumatology or epidemiology)

I have added “risk factor” as a keyword. The keywords are now “Diabetes, Frozen shoulder, Adhesive capsulitis, Risk factor, Meta-analysis”.

2. Abstract

*‘Onset’ word following ‘frozen shoulder’ is sometimes redundant. Please, avoid using ‘onset’ so many times, as using frozen shoulder is enough some of the times.

I have amended some sentences to avoid the use of the word “onset”.

3. Strengths

*Lines 39-43. Awkward wording. This sentence needs re-writing.

This sentence now reads “This systematic review is the first to summarise the results of studies estimating the longitudinal association between diabetes and the onset of frozen shoulder.”

4. Introduction

*Lines 8-9. Wrong way of citing, as Zeik is between brackets. Please, revise.

I have now changed all citations to be in the same style.

* The rationale and hypotheses need to be changed. Or at least, better explained.

I have now rewritten large parts of the introduction to better explain the rationale for, and the aim of, the study.

5. Methods

*Subheadings would allow a more fluent reading. Some subheadings such as search strategy, inclusion criteria, study selection, bias assessment, data analysis, etc., could be included.

The recommended subheadings have been added.

* Lines 44-45. It seems as if ALL cross-sectioned studies and case series found were studies where the full text was not available. And that this was the reason why they were excluded. Is this right?

This now reads "Cross-sectional studies and case series were excluded. Studies were also excluded if a full text could not be obtained."

* Was any assessment of heterogeneity done? Why not, as you mention that heterogeneity was a limitation of the study that preceded yours (i.e., Zeik's)

Yes, Section 2.5 explains that "Heterogeneity was assessed using Cochran's Q statistic, complemented by the I² index [26]."

* Was the same reviewer the one who reviewed all non-English language papers? It sounds strange.

This now reads "All non-English language papers were assessed by reviewers with appropriate language skills."

6. Results

* Page 7, line 3. 'Doing so' is a too informal term. The same happens with 'but specially so for...'

This now reads "Only one study [39] reported the types of diabetes that the participants had."

* Please reference the studies when you mention them. For instance: 'in one study (REF)... Frozen shoulder was identified using ICD-9 codes in three studies (REF)... This point applies to all Results section.

References have been added, as recommended.

* Page 7 lines 27-29. Past tense should be used instead of present tense.

This has been amended and now reads "...but the estimate for I^2 was imprecise as indicated by the wide 95% confidence interval. The influence analysis showed that excluding the largest study [37], which contained 4380 of the 5388 participants, greatly reduced the precision of the pooled estimate but did not substantially affect the value of the pooled estimate (Figure 3)."

7. Discussion

* Please, check the verb tenses.

This has now been amended.

* Page 8, line 13. Are the italics necessary?

The quote is no longer in italics.

* Figures: sometimes the thousands are separated by a comma and sometimes not: please, check it throughout the manuscript, including the tables as well.

Comma separators are now used for numbers equal to or greater than 10,000.

* Page 9, lines 13-15. Musculoskeletal pain is too vague for the conclusion, and you have not investigated this but frozen shoulder.

This now reads "Given the evidence in this review, clinicians should consider checking whether patients with diabetes are experiencing shoulder pain at their routine follow-up appointments."

7. Figures

* PRISMA flow diagram. Different letter sizes? (not sure)

This has been changed.

8. Appendices

* I would include Risk of Bias table in the manuscript instead of in the Appendix.

The table is now in the manuscript instead of the appendix.

REVIEWER 2 COMMENTS

This meta-analysis assesses the association between diabetes and frozen shoulder. I think the study was performed well in line with current guidelines for meta-analyses and do not have any major concerns.

I am not sure I agree with the statement that the funnel plot shows only slight asymmetry. The fact that the two studies on the bottom right are way over to the right is clear asymmetry, and in my opinion points to a significant publication risk even with the influence analysis stated. I would prefer to say there were too few studies to infer, than "slight asymmetry". Therefore, the authors might want to rephrase P7L56.

I agree that the plot shows asymmetry and have altered the text. The paragraph now states that "The funnel plot was unsymmetrical. However, given that a small number of studies were available, it was difficult to assess accurately whether any small-study bias was present or if the appearance was due

to chance. Since our influence analysis has shown that the inclusion/exclusion of any individual study had very little impact on the pooled effect estimate, any potential small-study bias would be unlikely to substantially affect the results.”

I think the main problem not addressed in the discussion is the diagnosis of “frozen shoulder”. As a co-author of one of the studies included [21] where every subject was examined and fulfilled the accurate definition of frozen shoulder, and as a clinician who sees patients with “frozen” shoulders frequently, I can guarantee we understand very little about the pathology. Obviously, some cases include an inflammatory process that has been reported in the subjects that have been biopsied. And the extreme cases that are “frozen” even under general anesthesia are definitely very hard to manage. In these cases, the other proposed mechanisms in P8L34-46 may be relevant. But the vast majority of subjects with “frozen shoulder” have no histopathological testing and have much milder signs and symptoms, which may not infer any intra-articular pathology. Many cases referred to me and my colleagues as “frozen shoulder” with that diagnosis encoded can be relieved in the clinic (at least temporarily) by various maneuvers. But they would be captured in “big data” studies like [24] & [25]. In the end, many of the cases are simply a clinical presentation, where the main manifestation is pain. In many, limited motion is probably secondary to the pain.

I do not think the authors necessarily want to delve into pain theory. But I think that the study is based on diagnoses of problematic reliability, made by a group of physicians whose training in MSK medicine may not be the best. This should be stated as a serious limitation.

I have added comments about frozen shoulder misclassification bias in the discussion.

“The two cohort studies in the review were both conducted using Electronic Health Records (EHRs). EHR datasets can provide large sample sizes with long follow-up periods and lots of covariate measurements [43]. Although, misdiagnosis and miscoding in EHRs will result in a risk of bias for frozen shoulder measurement [44]. Firstly, research in the UK [45] and in the Netherlands [46] has shown that general practitioners often use non-specific shoulder pain codes instead of codes for specific shoulder conditions, e.g., frozen shoulder. This would lead to an underdiagnosis of frozen shoulder. Further, this misclassification may be differential since clinicians may feel more confident in providing a specific frozen shoulder diagnosis in patients with diabetes due to the pre-existing knowledge of the association between the two conditions. Conversely, it has also been noted that frozen shoulder is sometimes used as a “waste-bin diagnosis” for patients presenting with any stiff and painful shoulder [47]. Thus, EHR data may include other shoulder conditions with similar clinical presentations being coded as frozen shoulder.”

REVIEWER 3 COMMENTS

This manuscript is well-written and presents results clearly. The authors point out that there was a lack of adjustment for confounders. What were the confounders? Does Table 1 give all the details of the confounders in the section “Variables conditioned on”. In some papers that seems very minimal for observational papers. Some discussion about the quality of the journals that accepted these very likely biased papers would enhance the manuscript.

The column “Variables conditioned on” in Table 1 does indeed include all covariates that have been conditioned on. I have commented on the risk of unaccounted confounding and how this affects the conclusions that can be drawn from the evidence, but I would not like to make further judgement about the quality of the journals publishing the research.

Should the PRISM flowchart be presented in the body of the manuscript rather than the appendix?

The PRISMA flowchart will appear as Figure 1 in the manuscript.

EDITOR(S) COMMENTS TO AUTHOR

- Please include the search dates and databases in the abstract.

These have now been included in the abstract.

- Please include, as a supplementary file, the precise, full search strategy (or strategies) for all databases, registers, and websites, including any filters and limits used.

All search strategies have now been included.

ASSOCIATE EDITOR COMMENTS

- The authors need to mention and discuss their other review (<https://pubmed.ncbi.nlm.nih.gov/34589691/>), which was a narrative synthesis of n=28 studies.

I have now referred to our other review in the introduction.

“Diabetes is a term used to describe a group of chronic diseases characterised by hyperglycaemia. The two most prevalent types of diabetes are type 1 and type 2 diabetes, making up 8% and 90% of cases, respectively [15]. It is well-known that people with diabetes are at risk of complications such as cardiovascular disease, retinopathy, neuropathy, and nephropathy [16]. Although the musculoskeletal complications of diabetes are not as well-known [17]. Musculoskeletal conditions, such as frozen shoulder, can significantly affect the quality of a patient’s life and should not be overlooked. Our previous systematic review and narrative synthesis of 28 studies has shown that patients with diabetes may also experience worse outcomes from frozen shoulder than people without frozen shoulder [18].”

- PRISMA 2020 is needed.

PRISMA 2020 has been used for the revised copy.

- The search strings for all databases are also missing.

Search strings are now included.

VERSION 2 – REVIEW

REVIEWER	Dueñas, Lirios University of Valencia, Department of physical therapy
REVIEW RETURNED	16-Oct-2022

GENERAL COMMENTS	The authors provided a response document that addresses the main areas of improvement that I noted with the original manuscript. Some specific comments are listed below. Addressing these comments will improve the clarity of the report. SPECIFIC COMMENTS Methods * The word ‘estimate’ in this sentence looks redundant. Please, use a synonym in order to get a clearer sentence. “(...) random-
---

	effects meta-analysis model was used to estimate a summary estimate when five or more studies were present”. Discussion * I miss, in the first paragraph of the discussion, which is the aim of the study, in order to refresh it and to start the discussion with. * The limitations are already settled, as proposed in my first review process, but I miss some words or sentence indicating that the limitations section is going to start (from the paragraph talking about the Electronic Health Records) * If you use the rule about using comma separators for numbers equal to or greater than 10,000, you should delete the comma in this sentence: ‘A cross-sectional study of 1,373 patients presenting with frozen shoulder estimated that the mean age of onset for frozen shoulder was 55.4 years with a standard deviation of 9.9’. *AGE’s or AGES?. You use both terms. Please, choose one, to give homogeneity. Tables *Table 1. I would write the word ‘studies’ in lowercase. ‘Characteristics of Studies on diabetes as a risk factor for frozen shoulder’.
--	--

REVIEWER	Finestone, Aharon Tel Aviv University
REVIEW RETURNED	07-Oct-2022

GENERAL COMMENTS	I have reviewed the updated manuscript, and the author’s response to the other reviewers and my own comments, and think they were answered appropriately, making the manuscript worthy of publication.
--

VERSION 2 – AUTHOR RESPONSE

SPECIFIC COMMENTS

Methods

* The word ‘estimate’ in this sentence looks redundant. Please, use a synonym in order to get a clearer sentence. “(...) *random-effects meta-analysis model was used to estimate a summary estimate when five or more studies were present*”.

This now reads “(...) random-effects meta-analysis model was used to calculate a summary estimate when five or more studies were present.”

Discussion

* I miss, in the first paragraph of the discussion, which is the aim of the study, in order to refresh it and to start the discussion with.

I have now added the following sentence at the start of the discussion paragraph.

“This systematic review aimed to summarise evidence from longitudinal observational studies to determine whether diabetes (types 1 and 2) is a risk factor for frozen shoulder.”

* The limitations are already settled, as proposed in my first review process, but I miss some words or sentence indicating that the limitations section is going to start (from the paragraph talking about the Electronic Health Records)

I have now added a sentence to indicate that the limitations section is going to start.

“The following three paragraphs describe some limitations that may complicate the understanding of the association between diabetes and the onset of frozen shoulder.”

* If you use the rule about using comma separators for numbers equal to or greater than 10,000, you should delete the comma in this sentence: ‘*A cross-sectional study of 1,373 patients presenting with frozen shoulder estimated that the mean age of onset for frozen shoulder was 55.4 years with a standard deviation of 9.9.*

The comma in “1,373” has now been removed.

*AGE’s or AGEs?. You use both terms. Please, choose one, to give homogeneity.

“AGEs” is now used throughout.

Tables

*Table 1. I would write the word ‘studies’ in lowercase. ‘*Characteristics of Studies on diabetes as a risk factor for frozen shoulder.*

The word “studies” is now in lowercase.